## Research Article

functional impairment; psychological distress; depression diagnosis; older adults; BRFSS

**Corresponding author:**
Sunkanmi Folorunsho;
Email: sfolorunsho2@huskers.unl.edu

# Functional impairment without depression diagnosis as an indicator of psychological distress in older adults

Sunkanmi Folorunsho[1] [iD], Olabisi Promise Lawal[2], Ndidiamaka Christiana Ani[3] and Michael Ameyaw Somuah[4]

[1]Sociology, University of Nebraska–Lincoln, USA; [2]Medical Laboratory Science, University of Benin, Nigeria; [3]Jack, Joseph and Morton Mandel School of Applied Social Sciences, Case Western Reserve University, USA and [4]Department of Economics, The George Washington University, USA

## Abstract

Psychological distress can occur even without a depression diagnosis. Many older adults have functional limitations that hinder daily activities, yet their emotional needs often go unrecognized. This study examined whether functional impairment is associated with psychological distress in older adults and whether this relationship varies by depression-diagnosis status. Data came from the 2023 Behavioral Risk Factor Surveillance System for U.S. adults aged 65 and older (N = 95,325). Functional impairment was defined as having 14 or more days in the past month when poor health limited usual activities. Psychological distress was measured by days of poor mental health and a binary indicator of high distress. Survey-weighted regression analyses tested main and interaction effects of functional impairment and depression diagnosis while adjusting for sociodemographic and behavioral factors. Functional impairment was linked to greater distress. Predicted estimates showed the highest distress among those with both impairment and a depression diagnosis (about 11 poor mental health days). Those with impairment only averaged about 6 days, those with a diagnosis only about 8, and those with neither condition about 3. Functional impairment may reveal hidden distress in older adults without diagnosed depression. Adding physical-function indicators to screening could help identify vulnerable individuals earlier.

## Impact statement

This study identifies functional impairment as a potential behavioral marker of hidden psychological distress in older adults. Using nationally representative data, we demonstrate that even in the absence of a depression diagnosis, older adults with functional limitations experience significantly higher levels of psychological distress. These findings challenge the reliance on clinical diagnosis as the sole gateway to mental health care and underline the need for more inclusive screening approaches that incorporate physical functioning as an early signal of psychological vulnerability. Integrating functional assessments into primary care and public health surveillance could help identify older adults whose distress remains unrecognized or untreated. The results also highlight the importance of addressing socioeconomic disparities that compound the burden of both impairment and mental health risk. By linking physical limitation to psychological well-being, this study advances the Stress Process framework and offers actionable guidance for building holistic, equitable and preventive models of care for aging populations.

## Introduction

Functional impairment is a prevalent and often progressive condition among older adults, reflecting underlying physical, cognitive and psychological limitations. It commonly arises from age-related physiological decline, multimorbidity and diminished resilience (Freedman et al., 2016; Amlak et al., 2025). Manifestations range from difficulty performing basic activities of daily living, such as bathing and dressing, to challenges with complex instrumental tasks like managing finances or medications. In the United States, about one in four adults aged 65 years and older experiences some form of functional limitation (Centers for Disease Control and Prevention, 2023). With increasing longevity, functional impairment is expected to become even more widespread and to exert a growing impact on population health and well-being.

Functional impairment in later life has been consistently linked to reduced quality of life, greater risk of institutionalization and increased psychological distress (Mayer et al., 2021; Cano

et al., 2024). Psychological distress in this study refers to recurrent experiences of negative affect such as stress, anxiety or low mood as well as related emotional strain that signals reduced mental well-being even when they do not meet diagnostic thresholds for a depressive disorder (Mirowsky and Ross, 2002; Folorunsho, 2025). Among these consequences, psychological distress is particularly common, as limitations in physical functioning often restrict social participation, autonomy and role fulfillment, leading to feelings of frustration, isolation and emotional strain (Folorunsho and Okyere, 2025; Islam et al., 2025). Although previous research has documented strong associations between disability and depression (Rodda et al., 2011; Taylor, 2023), fewer studies have examined how functional impairment relates to psychological distress among those without a clinical diagnosis of depression. Even when they do not meet diagnostic criteria, many older adults experience frequent days of poor mental health that reflect subclinical distress with tangible effects on health and daily functioning (Mojtabai and Olfson, 2004; Bentur and Heymann, 2020).

Older adults who lack a depression diagnosis may be excluded from formal care systems despite substantial emotional needs. Functional impairment may therefore act as a behavioral marker of hidden distress, revealing unmet psychological vulnerability among aging populations (Goodarzi et al., 2024). This concern is heightened by persistent barriers to mental health care, including stigma, cost and limited access to qualified providers (Gonzalez et al., 2010). Understanding whether functional impairment independently predicts distress among those without diagnosed depression is critical for identifying underserved groups and improving prevention strategies.

Population health surveillance systems such as the Behavioral Risk Factor Surveillance System (BRFSS) offer a valuable opportunity to investigate this relationship using nationally representative data. The BRFSS collects standardized measures of functional limitation, mental health and depression diagnosis across diverse demographic and behavioral profiles. Using these data, the present study aimed to determine whether functional impairment is independently associated with psychological distress among older adults in the United States and whether this association varies by depression-diagnosis status. Clarifying these relationships can enhance understanding of how physical limitations and mental health intersect in late life and inform the development of more inclusive and proactive mental health screening approaches.

## Theoretical framework

This study is grounded in the Stress Process Model developed by Pearlin and colleagues, which has become one of the most influential frameworks for understanding the social origins of stress and mental health outcomes across the life course (Pearlin et al., 1981; Pearlin, 1989). The model proposes that stressors, defined as chronic strains or life events that threaten or disrupt normal functioning, can trigger adverse psychological and health outcomes. The effects of these stressors are shaped by the availability of coping resources such as social support, self-esteem, mastery and other structural or personal factors that may buffer or exacerbate distress (Thoits, 2010; Aneshensel and Avison, 2015). In this view, mental health is not only a product of individual characteristics but also of the structural conditions that expose people to unequal stress burdens and influence their ability to respond effectively to them.

Subsequent extensions of the Stress Process Model have emphasized the dynamic and cumulative nature of stress over time. Pearlin (2009) describes stress proliferation as the process by which one stressor can give rise to others, producing a cascade of role strains and emotional demands that intensify psychological vulnerability. For older adults, chronic physical conditions such as functional impairment represent enduring stressors that can erode a sense of control, autonomy and social participation (George, 2007; Turner et al., 2016). When functional impairment limits daily activities, it may generate both primary stress from the loss of independence and secondary stress from social withdrawal or role disruption, thereby increasing the risk of sustained emotional strain and psychological distress. Additionally, integrating a life course perspective further enriches this framework by situating late-life stress within the accumulation of earlier experiences and exposures. Over time, disadvantages related to socioeconomic position, health and access to resources can accumulate, leading to weathering and diminished coping reserves in later life (Taylor and Seeman, 1999; Pearlin, 2009). From this perspective, functional impairment in older age may not only reflect immediate physical limitations but also the cumulative consequences of lifelong inequalities that make individuals more susceptible to psychological distress.

In applying this theoretical foundation to the current study, functional impairment is conceptualized as the primary stressor, representing a chronic condition that restricts daily functioning and disrupts engagement in valued activities. Psychological distress, measured as the number of poor mental health days, represents the emotional consequence of this strain. Depression diagnosis is not treated as a stressor but rather as a clinical indicator of recognized mental health burden that may modify the relationship between impairment and distress. Specifically, the model suggests that functional impairment should elevate distress even among those without a depression diagnosis, while the coexistence of both impairment and diagnosis should produce the highest levels of distress because of the combined effects of physical and psychological strain.

## Conceptual model

Figure 1 presents the conceptual model for the study, grounded in the Stress Process Model (Pearlin et al., 1981). The framework illustrates that functional impairment acts as the primary stressor that directly influences psychological distress, representing the emotional outcome of chronic strain in late life. Depression diagnosis is positioned as a moderating variable that alters the strength or direction of this relationship. The solid arrow indicates the direct pathway between functional impairment and psychological distress, while the dashed line depicts the moderating influence of depression diagnosis on this relationship. The model suggests that

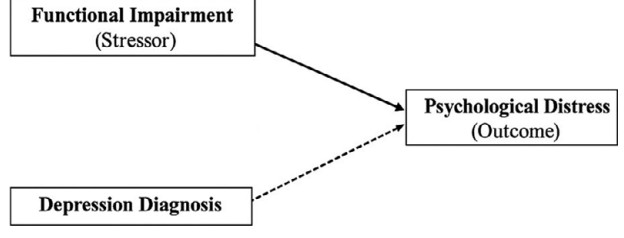

**Figure 1.** Conceptual model of study.

psychological distress is expected to be highest among older adults who experience both functional impairment and a depression diagnosis, intermediate among those with impairment alone, and lowest among those without either condition. This conceptual structure captures the hypothesized interactive and additive pathways linking physical limitation, clinical status and mental health outcomes in older adulthood.

## Measures

### Independent variable

#### Functional impairment

This was assessed using the BRFSS item, "During the past 30 days, for about how many days did poor physical or mental health keep you from doing your usual activities, such as self-care, work or recreation?" (CDC, 2023). Responses ranged from 0 to 30 days. Following established conventions in public health research, respondents reporting 14 or more days of activity limitation were categorized as functionally impaired. This threshold is widely used to denote serious functional disruption and clinically meaningful declines in daily functioning (Gerbi et al., 2016; Cree et al., 2017).

### Dependent variables

#### Psychological distress

Two measures of psychological distress were examined. The first was a continuous variable representing the number of days in the past 30 that participants reported their mental health was "not good," capturing symptoms such as stress, anxiety or emotional distress. The second measure was a binary indicator of high psychological distress, defined as reporting 14 or more days of poor mental health during the previous month. This cutoff reflects clinically significant distress and has been validated as a population-based indicator of mental health burden (Strine et al., 2008; Cree et al., 2017).

### Moderating variable

#### Depression diagnosis

This was determined from participants' responses to the question, "Has a doctor or other health professional ever told you that you have a depressive disorder, including depression, major depression, dysthymia or minor depression?" Responses were coded dichotomously (yes/no). This single-item measure has been widely applied in BRFSS studies to distinguish diagnosed depressive disorders from self-reported psychological distress (Strine et al., 2006; Han et al., 2021).

### Covariates

Covariates were selected based on previous literature linking sociodemographic and health factors to mental health outcomes in older adults. Age was treated as a continuous variable in years, and sex was coded as male or female. Educational attainment was categorized as less than high school, high school graduate, some college or college graduate. Household income was grouped into four categories: less than $25,000, $25,000–$49,999, $50,000–$74,999 and $75,000 or more. Insurance status was classified as private, public (e.g., Medicare or Medicaid) or uninsured, reflecting established

associations between insurance type, service access and diagnostic likelihood (Sivakumar et al., 2020; Lee et al., 2022).

Two behavioral health variables were also included. Physical activity was measured by whether participants engaged in any non-work-related physical exercise or activity in the past 30 days (yes/no). Routine medical check-up was coded as "within the past year" or "one year ago or more," given evidence that delayed preventive care is associated with underdiagnosed mental health conditions and poorer late-life outcomes. All covariates were derived from the 2023 BRFSS and coded in accordance with CDC analytic guidelines.

### Statistical analysis

Analyses were conducted in four stages. First, descriptive statistics were generated to compare sociodemographic, behavioral and health characteristics across groups defined by functional impairment and depression-diagnosis status (Tables 1 and 2). Weighted means and proportions were calculated, and group differences were evaluated to identify significant variations across key characteristics. Second, multivariable linear regression models were estimated to examine the association between functional impairment and the number of poor mental health days. Third, logistic regression models were used to estimate the odds of high psychological distress, defined as reporting ≥14 poor mental health days in the past 30 days. Each modeling sequence proceeded in four steps: (1) testing the main effect of functional impairment, (2) adding depression diagnosis, (3) including the interaction term between functional impairment and depression diagnosis and (4) adjusting for all covariates. To facilitate interpretation of the interaction, predicted marginal means of poor mental health days were computed for subgroups defined by functional impairment and depression-diagnosis status. These predicted values were visualized in a line plot (Figure 2) to illustrate variations in psychological distress by functional and diagnostic status. All analyses were performed in Stata 18.0 (StataCorp LLC, College Station, TX). Survey weights, clustering and strata adjustments were applied to account for the complex sampling design of the BRFSS. Statistical significance was defined as $p < .05$ (two-tailed).

## Results

### Sample characteristics

Descriptive characteristics of the analytic sample are summarized in Tables 1 and 2. The mean age of respondents was 72.5 years (SD = 9.3; range = 65–99), and 57.8% were female. Approximately one-quarter (23.4%) reported functional impairment, while one in five (20.5%) reported having been diagnosed with depression. On average, participants reported 6.2 poor mental health days in the past month, reflecting moderate psychological distress in this older population. As shown in Table 1, significant sociodemographic and behavioral differences emerged by impairment status. Older adults with functional impairment were, on average, older (74.2 vs. 71.4 years, $p < .001$) and more likely to be female (65.7% vs. 52.1%, $p < .001$). They also had lower educational attainment and household income compared to those without impairment (both $p < .05$). Impaired individuals were less likely to engage in physical activity (51.2% vs. 69.1%, $p < .001$) and less likely to have had a recent medical check-up (68.1% vs. 79.4%, $p < .001$). Psychologically, the burden was substantial: 39.4% of those with impairment experienced

**Table 1.** Descriptive statistics stratified by functional impairment status

| Variable | Not impaired | Impaired | *p*-value |
|---|---|---|---|
| Age (mean ± SE) | 71.4 ± 0.3 | 74.2 ± 0.4 | <0.001 |
| Sex (%) | | | |
| Female | 52.1 | 65.7 | <0.001 |
| Male | 47.9 | 34.3 | <0.001 |
| Education level (%) | | | |
| Less than high school | 12.3 | 20.4 | <0.01 |
| High school graduate | 29.4 | 33.5 | 0.03 |
| Some college | 29.9 | 22.1 | <0.01 |
| College graduate | 28.4 | 24.0 | 0.04 |
| Household income (%) | | | |
| <$25,000 | 21.2 | 37.3 | <0.001 |
| $25,000–$49,999 | 31.0 | 28.1 | 0.08 |
| $50,000–$74,999 | 24.8 | 17.4 | 0.01 |
| ≥$75,000 | 23.0 | 17.2 | 0.04 |
| Insurance type (%) | | | |
| Public insurance | 60.5 | 66.7 | 0.05 |
| Private insurance | 35.0 | 29.5 | 0.04 |
| Uninsured | 4.5 | 3.8 | 0.22 |
| Physical activity (past 30 days) (%) | | | |
| Yes | 69.1 | 51.2 | <0.001 |
| No | 30.9 | 48.8 | <0.001 |
| Routine medical check-up (%) | | | |
| Within the past year | 79.4 | 68.1 | <0.001 |
| 1+ years ago | 20.6 | 31.9 | <0.001 |
| Depression diagnosis (%) | | | |
| Yes | 15.3 | 28.9 | <0.001 |
| No | 84.7 | 71.1 | <0.001 |
| High psychological distress (≥14 days poor MH) (%) | 11.5 | 39.4 | <0.001 |
| Mean days mental health "Not Good" (± SE) | 4.1 ± 0.3 | 11.8 ± 0.5 | <0.001 |

*Note:* Percentages and means are weighted using BRFSS complex sampling design weights (2023 dataset).

**Table 2.** Descriptive statistics stratified by depression diagnosis

| Variable | No depression diagnosis | With depression diagnosis | *p*-value |
|---|---|---|---|
| Age (mean ± SE) | 72.7 ± 0.3 | 71.1 ± 0.4 | <0.001 |
| Sex (%) | | | |
| Female | 53.2 | 66.9 | <0.001 |
| Male | 46.8 | 33.1 | <0.001 |
| Education level (%) | | | |
| Less than high school | 13.5 | 18.0 | 0.01 |
| High school graduate | 31.2 | 28.5 | 0.04 |
| Some college | 27.4 | 27.9 | 0.88 |
| College graduate | 27.9 | 25.6 | 0.03 |
| Household income (%) | | | |
| <$25,000 | 23.5 | 39.1 | <0.001 |
| $25,000–$49,999 | 30.8 | 29.1 | 0.31 |
| $50,000–$74,999 | 23.6 | 18.0 | 0.02 |
| ≥$75,000 | 22.1 | 13.8 | <0.001 |
| Insurance type (%) | | | |
| Public insurance | 60.7 | 65.2 | 0.04 |
| Private insurance | 34.2 | 30.3 | 0.05 |
| Uninsured | 5.1 | 4.5 | 0.33 |
| Physical activity (past 30 days) (%) | | | |
| Yes | 66.9 | 52.0 | <0.001 |
| No | 33.1 | 48.0 | <0.001 |
| Routine medical check-up (%) | | | |
| Within the past year | 78.6 | 72.4 | 0.002 |
| 1+ years ago | 21.4 | 27.6 | 0.002 |
| Functional impairment (≥14 days poor health) (%) | 22.3 | 47.8 | <0.001 |
| High psychological distress (≥14 days poor MH) (%) | 10.4 | 45.1 | <0.001 |
| Mean days mental health "Not Good" (± SE) | 3.8 ± 0.3 | 13.1 ± 0.5 | <0.001 |

*Note:* All estimates are weighted using BRFSS complex survey design for 2023 data.

high psychological distress (≥14 poor mental health days), compared to 11.5% among those without impairment (*p* < .001). Differences were also observed by depression diagnosis (Table 2). Respondents with a depression diagnosis were younger on average (71.1 vs. 72.7 years, *p* < .001) and more likely to be female (66.9% vs. 53.2%, *p* < .001). They were also more likely to report low income and limited physical activity (both *p* < .001). Notably, nearly half of those with depression (47.8%) reported functional impairment, and they experienced far more poor mental health days than their non-diagnosed counterparts (13.1 vs. 3.8, *p* < .001). Together, these descriptive results indicate that both functional impairment and depression diagnosis cluster within socioeconomically disadvantaged and physically inactive subgroups, highlighting overlapping vulnerabilities among older adults.

### Association between functional impairment and psychological distress

Results from the multivariable linear regression models predicting the number of poor mental health days are presented in Table 3. In the unadjusted model (Model 1), functional impairment was associated with a 3.12-day increase in poor mental health per month (*p* < .001). This association persisted, although attenuated, after adjusting for depression diagnosis and covariates. In the fully adjusted model (Model 4), older adults with functional impairment reported an average of 1.87 additional poor mental health days compared with those without impairment (*p* < .001). Depression diagnosis remained the strongest predictor of distress, associated with 4.93 more poor mental health days (*p* < .001). The interaction

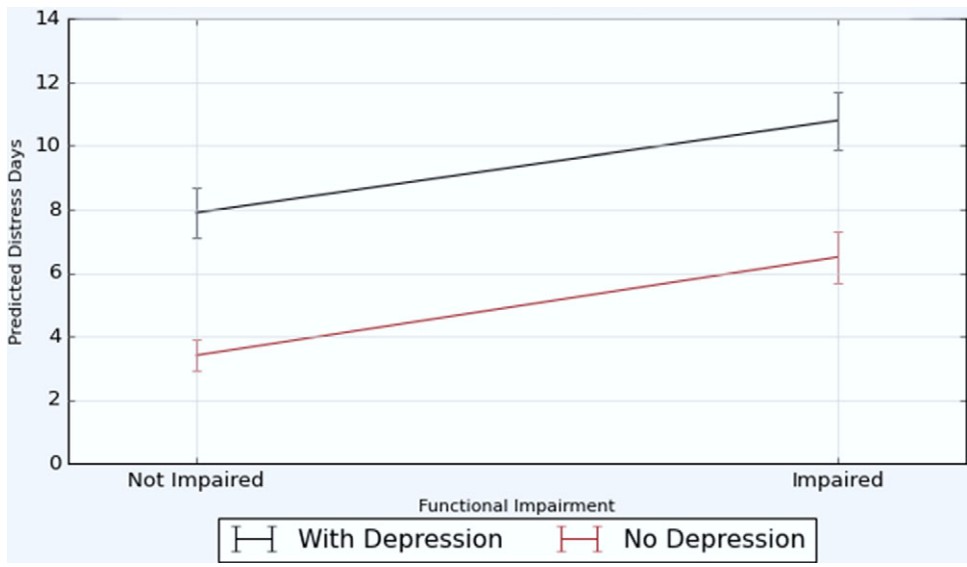

**Figure 2.** Predicted poor mental health days by impairment and depression.

**Table 3.** Multivariable linear regression predicting days of poor mental health

| Variable | Model 1 *b* (SE) | Model 2 *b* (SE) | Model 3 *b* (SE) | Model 4 *b* (SE) |
|---|---|---|---|---|
| Functional impairment (≥14 days poor health) | 3.12 (0.25)*** | 2.64 (0.26)*** | 2.21 (0.27)*** | 1.87 (0.29)*** |
| Depression diagnosis (yes) | — | 5.33 (0.32)*** | 5.20 (0.33)*** | 4.93 (0.36)*** |
| Impairment × no depression diagnosis | — | — | 1.48 (0.42)** | 1.35 (0.41)** |
| Age (years) | — | — | — | −0.07 (0.01)*** |
| Female (ref: male) | — | — | — | 0.68 (0.22)** |
| Education (ref: college graduate) | | | | |
| Less than high school | — | — | — | 1.05 (0.41)* |
| High school graduate | — | — | — | 0.66 (0.33)* |
| Some college | — | — | — | 0.44 (0.29) |
| Income (ref: ≥$75,000) | | | | |
| <$25,000 | — | — | — | 2.71 (0.38)*** |
| $25,000–$49,999 | — | — | — | 1.39 (0.32)*** |
| $50,000–$74,999 | — | — | — | 0.84 (0.30)** |
| Insurance (ref: private) | | | | |
| Public | — | — | — | 0.41 (0.27) |
| Uninsured | — | — | — | 0.93 (0.61) |
| No physical activity | — | — | — | 2.88 (0.31)*** |
| No recent check-up (≥1 year ago) | — | — | — | 1.25 (0.26)*** |

*Note:* Outcome = Days of poor mental health in the past 30 days. Unstandardized regression coefficients reported; standard errors in parentheses. *p < 0.05, **p < 0.01, ***p < 0.001.

between functional impairment and absence of depression diagnosis was statistically significant (β = 1.35, *p* < .01), suggesting that impairment independently predicted distress even among those without a depression diagnosis. In other words, older adults experiencing functional limitations, even without formal recognition of depression, displayed elevated psychological distress relative to their unimpaired peers. Sociodemographic covariates were also significantly associated with distress. Women, individuals with lower education or income, and those who were physically inactive

reported more days of poor mental health, while older age was linked to slightly fewer distress days.

### Predictors of high psychological distress

Results from the logistic regression models predicting high psychological distress (≥14 days of poor mental health) are shown in Table 4. Consistent with the linear regression findings, functional impairment was strongly associated with elevated odds of high

**Table 4.** Logistic regression predicting high psychological distress (≥14 days poor mental health)

| Variable | Model 1 OR (95% CI) | Model 2 OR (95% CI) | Model 3 OR (95% CI) | Model 3 OR (95% CI) |
|---|---|---|---|---|
| Functional impairment (≥14 days poor health) | 3.42 (2.91–4.01)*** | 2.81 (2.37–3.34)*** | 2.34 (1.98–2.78)*** | 2.08 (1.72–2.51)*** |
| Depression diagnosis (yes) | — | 6.20 (5.33–7.21)*** | 6.01 (5.14–7.02)*** | 5.66 (4.73–6.77)*** |
| Impairment × no depression diagnosis | — | — | 1.95 (1.34–2.85)** | 1.72 (1.18–2.51)** |
| Age (years) | — | — | — | 0.96 (0.95–0.97)*** |
| Female (ref: male) | — | — | — | 1.43 (1.20–1.70)*** |
| Education (ref: college graduate) | | | | |
| Less than high school | — | — | — | 1.62 (1.17–2.24)** |
| High school graduate | — | — | — | 1.38 (1.04–1.83)* |
| Some college | — | — | — | 1.24 (0.95–1.61) |
| Income (ref: ≥$75,000) | | | | |
| <$25,000 | — | — | — | 2.94 (2.13–4.06)*** |
| $25,000–$49,999 | — | — | — | 1.91 (1.41–2.59)*** |
| $50,000–$74,999 | — | — | — | 1.46 (1.07–2.00)* |
| Insurance (ref: private) | | | | |
| Public | — | — | — | 1.18 (0.91–1.54) |
| Uninsured | — | — | — | 1.52 (0.87–2.65) |
| No physical activity | — | — | — | 3.09 (2.51–3.81)*** |
| No recent check-up (≥1 year ago) | — | — | — | 1.55 (1.27–1.88)*** |

*Note:* Outcome = High psychological distress (≥14 days poor mental health). Odds ratios (OR) with 95% confidence intervals reported. *p < 0.05, **p < 0.01, ***p < 0.001.

distress. In the unadjusted model, impaired older adults were more than three times as likely to experience high psychological distress compared with those without impairment (OR = 3.42, 95% CI [2.91–4.01], *p* < .001). This association remained significant, although attenuated, after the inclusion of depression diagnosis and all covariates. In the fully adjusted model, functional impairment was associated with twice the odds of high psychological distress (OR = 2.08, 95% CI [1.72–2.51], *p* < .001). Depression diagnosis was the strongest independent predictor in all models (OR = 5.66, 95% CI [4.73–6.77], *p* < .001), indicating that diagnosed individuals were substantially more likely to experience frequent mental distress. The significant interaction between functional impairment and the absence of depression diagnosis (OR = 1.72, 95% CI [1.18–2.51], *p* < .01) further suggested that functional limitations predicted high distress even among those who had not received a clinical diagnosis.

### Predicted marginal estimates and interaction effects

Predicted marginal means from the fully adjusted model are presented in Table 5 and illustrated in Figure 2. The highest predicted number of poor mental health days was observed among older adults with both functional impairment and a depression diagnosis. Those with only a depression diagnosis reported fewer poor mental health days, followed by those with impairment but no diagnosis. Participants with neither condition reported the fewest poor mental health days overall.

Figure 2 shows that psychological distress increased consistently with functional impairment, regardless of diagnostic status. The slope was steeper among individuals without a depression diagnosis, indicating that functional impairment may impose a greater emotional burden on those whose symptoms are not formally

**Table 5.** Predicted marginal number of poor mental health days by functional impairment and depression-diagnosis status (n = 95,325)

| Functional impairment | Depression diagnosis | Predicted distress days (95% CI) |
|---|---|---|
| Not impaired | No | 3.4 (2.9–3.9) |
| Not impaired | Yes | 7.9 (7.1–8.7) |
| Impaired | No | 6.5 (5.7–7.3) |
| Impaired | Yes | 10.8 (9.9–11.7) |

*Note:* Predicted values and 95% confidence intervals were estimated using the fully adjusted multivariable linear regression model (Model 4, Table 3). Estimates are adjusted for age, sex, education, household income, insurance type, physical activity and time since last routine check-up.

recognized. This pattern is consistent with the Stress Process Model, which conceptualizes functional limitations as chronic stressors that heighten vulnerability to psychological distress, particularly when mental health needs are unmet.

### Discussion

This study examined how functional impairment relates to psychological distress among older adults and explored whether this relationship varies by depression diagnosis. Both functional impairment and depression diagnosis were independently associated with greater distress, and their combination intensified psychological burden. Importantly, even in the absence of a formal depression diagnosis, functional impairment remained a significant predictor of psychological distress, underlining the potential of impairment as a behavioral marker for unrecognized or subclinical emotional difficulties. These findings contribute to a growing

literature documenting the interdependence of physical and mental health in aging populations. Research has consistently shown that functional limitations predict depressive symptoms, anxiety and general psychological distress in older adults (Choi and Lee, 2021; Cano et al., 2024; Wang et al., 2024). The loss of physical autonomy often disrupts daily routines, reduces opportunities for social interaction and increases dependency on others for basic needs. Such disruptions can undermine an individual's sense of control and self-efficacy, central pillars of psychological well-being. Within the Stress Process Model (Pearlin et al., 1981), functional impairment represents a chronic stressor that initiates secondary strains such as social isolation, role loss or perceived uselessness, ultimately culminating in heightened emotional distress.

The persistence of this relationship among those without a depression diagnosis raises critical questions about the recognition and diagnosis of mental health conditions in older adults. Older adults may normalize emotional distress as a natural part of aging or attribute it to physical illness, resulting in underreporting of psychological symptoms. Clinicians, in turn, may prioritize physical complaints and overlook the psychological sequelae of impairment (Lutz and Van Orden, 2020; Ramírez López et al., 2024). These diagnostic gaps are further compounded by barriers such as stigma, limited mobility and financial constraints that restrict access to mental health services (Conner et al., 2010). Consequently, functionally impaired individuals without a formal diagnosis may represent an at-risk group whose psychological needs remain unmet.

From a theoretical standpoint, the observed additive effects of impairment and depression align with the concept of stress proliferation (Newland et al., 2013). This framework posits that stressors in one domain of life can give rise to additional stressors across others, producing cumulative emotional consequences. The finding that those with both impairment and depression reported more than 10 poor mental health days per month reflects this cumulative load. Functionally impaired older adults may experience compounding stress from physical limitations, diminished independence, financial strain and weakened social ties. This syndemic pattern, where physical and psychological adversities reinforce each other, highlights the need for integrative approaches to geriatric care that move beyond single-diagnosis treatment models (Marengoni et al., 2011).

The interaction effects observed in this study are also noteworthy. Even after adjusting for sociodemographic and behavioral factors, impairment remained strongly associated with distress among individuals without a depression diagnosis. This supports the hypothesis that functional limitations can act as early indicators of unrecognized psychological strain. Figure 2 illustrates this pattern, showing that the emotional burden of impairment persists regardless of clinical diagnosis, although the effect is more pronounced among those not formally diagnosed. In practical terms, this suggests that functional screening could be a valuable entry point for mental health assessment in primary care and community settings. For instance, older adults reporting prolonged activity limitations might benefit from brief psychological evaluations even in the absence of self-reported depressive symptoms.

Beyond the psychological implications, the findings reveal striking social gradients. Older adults with lower income, less education and public insurance were disproportionately represented among those with both functional impairment and depression. These results mirror existing evidence linking socioeconomic disadvantage to both physical and mental health disparities in late life (Lee et al., 2022). Limited financial resources may restrict access

to rehabilitative care, assistive devices or environments that support independence. Similarly, older adults living in economically disadvantaged neighborhoods may have fewer opportunities for social engagement, leading to isolation and distress. Within the Stress Process framework, these social inequalities represent structural stressors that intensify the effects of individual impairment. Addressing them requires system-level policies that expand access to preventive health services, community-based rehabilitation and mental health support for aging populations.

The present findings have several implications for public health practice and clinical care. First, they point to the importance of adopting a more holistic and proactive approach to mental health screening among older adults. Routine assessment of functional status could serve as a low-cost, non-stigmatizing strategy for identifying those at risk of psychological distress. Second, interventions designed to maintain or restore functional ability, such as physical therapy, adaptive technologies and exercise programs, may have secondary benefits for emotional well-being. Third, the results point to the need for integrated care models that bridge primary, mental and rehabilitative health services. Collaborative interventions that combine physical rehabilitation with psychosocial support have been shown to reduce depressive symptoms and improve quality of life in aging populations (González-González et al., 2022).

This study has several strengths. It draws on a large, nationally representative dataset and employs survey-weighted analyses to enhance population-level generalizability. The use of both continuous and categorical measures of distress allows for a nuanced understanding of mental health outcomes. Furthermore, by examining interaction effects and predicted marginal estimates, this study moves beyond simple associations to highlight the complexity of co-occurring health challenges in later life. The analysis extends previous research by empirically demonstrating that distress is elevated even among functionally impaired older adults without formal diagnoses, a group that may otherwise remain invisible in surveillance and intervention efforts. Nevertheless, several limitations should be noted. The data are cross-sectional, precluding causal inference. All measures are self-reported, which may introduce recall bias or social desirability effects. The BRFSS depression measure is based on lifetime diagnosis rather than current clinical status, and the functional impairment measure, although widely used, does not capture specific dimensions of limitation such as mobility or ADLs. Additionally, the dataset lacks contextual variables that may mediate or moderate these relationships, including caregiving arrangements, social networks or healthcare utilization patterns. Future research should employ longitudinal designs and multidimensional measures of functioning to examine causal pathways and identify modifiable intervention points. Finally, while these findings are derived from U.S. data, the mechanisms linking functional impairment and psychological distress are likely relevant in other cultural and economic contexts. However, the strength and expression of these associations may differ according to cultural beliefs about aging, family support structures and accessibility of mental health services. Comparative studies across regions and income levels could illuminate how social and policy environments shape the intersection between physical and mental health in later life.

## Conclusion

Functional impairment signifies more than a physical limitation; it represents a marker of emotional vulnerability that is often

overlooked in older adults. The strong and persistent association between impairment and distress, even among those without a depression diagnosis, pointing to the need to expand mental health screening and intervention frameworks beyond clinical boundaries. Integrating assessments of functional status into routine care may help detect hidden distress and provide early opportunities for intervention. As populations continue to age, ensuring that physical health, psychological well-being and social support are addressed in a unified manner will be essential for promoting healthy and dignified aging.

**Open peer review.** To view the open peer review materials for this article, please visit http://doi.org/10.1017/gmh.2025.10098.

**Data availability statement.** The data that support the findings of this study are publicly available from the U.S. Centers for Disease Control and Prevention (CDC) Behavioral Risk Factor Surveillance System (BRFSS) repository at https://www.cdc.gov/brfss/.

**Acknowledgments.** The authors thank the CDC for providing access to the BRFSS data. No additional acknowledgments are applicable.

**Author contribution.** S.F. conceptualized the study, performed the statistical analyses and led the writing of the manuscript. O.P.L. contributed to the literature review, data interpretation and manuscript revision. N.C.A. assisted with theoretical framing and the synthesis of related empirical literature. M.A.S. contributed to the data management and provided critical feedback on the analytical approach. D.E. provided senior oversight, conceptual guidance and substantive revisions to improve intellectual and methodological clarity. All authors reviewed and approved the final version of the manuscript.

**Financial support.** This research received no specific grant from any funding agency, commercial or not-for-profit

**Competing interests.** The authors declare no conflict of interest.

**Ethics statement.** This study involved secondary analysis of publicly available, deidentified data from the 2023 Behavioral Risk Factor Surveillance System (BRFSS). As such, it was exempt from institutional review board (IRB) approval.

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
