## [Reviewer Report]

The article is dealing with an interesting topic in the psychogeriatric’s field. The data pool seems to have interesting info to provide. This is a well-designed study, with good statistical analysis in a very challenging topic. Some aspects could be ameliorated in order to improve the present manuscript.

The title of this paper “Functional Impairment without Depression Diagnosis as a Behavioral Marker of Cognitive Decline in Older Adults” does not clearly describe the scope of the current study. The title speaks about “cognitive decline”, but in the content there is no such report, measure or assessment scale relevant to cognitive function.

The Introduction part would be more completed if the meaning of the term “psychological distress” was included.

In the Conceptual Model section, the phrase “Depression diagnosis is included as a clinical indicator that reflects the recognized mental health burden” needs more explanation. Also, in Figure 1, the arrows should be changed or reversed in order to better reflect the meaning the authors wish.

In the Methods section, it is not very clear in which groups the patients are assigned.

The Discussion part is well illustrative and the strengths and especially the limitations of the study are veritably reported.

---

## [Reviewer Report]

Thank you for your excellent and important paper. There is an optional module on subjective cognitive decline included in BRFSS that some states ask, some states don’t every year (and in different versions of BRFSS). The term “cognitive decline” refers to a gradual decline in cognitive abilities, such as memory, attention, reasoning, and decision-making. Your title includes the term “cognitive decline,” but your paper throughout rightly talks about “cognitive vulnerability” and “psychological distress.” All I would recommend is that you do not use the term “cognitive decline” in the title, and instead either use “cognitive vulnerability” or “psychological distress.”